# Maternal-Child Health Outcomes from Pre- to Post-Implementation of a Trauma-Informed Care Initiative in the Prenatal Care Setting: A Retrospective Study

**DOI:** 10.3390/children8111061

**Published:** 2021-11-18

**Authors:** Nicole Racine, Whitney Ereyi-Osas, Teresa Killam, Sheila McDonald, Sheri Madigan

**Affiliations:** 1Department of Psychology, University of Calgary, Calgary, AB T2N 1N4, Canada; nicole.racine2@ucalgary.ca; 2Cumming School of Medicine, University of Calgary, Calgary, AB T2N 4N1, Canada; whitney.ereyiosas@ucalgary.ca (W.E.-O.); teresa.killam@me.com (T.K.); 3Riley Park Maternity Clinic, Calgary, AB T2N 1B9, Canada; 4Department of Community Health Sciences, Cumming School of Medicine, University of Calgary, Calgary, AB T2N 4Z6, Canada; sheila.mcdonald@albertahealthservices.ca; 5Maternal Child Health, Department of Research and Innovation, Population, Public and Aboriginal Health, Alberta Health Services, Calgary, AB T2W 3N2, Canada

**Keywords:** adverse childhood experiences, screening, trauma-informed care, health, pregnancy

## Abstract

Background: There has been an increase in use of trauma-informed care (TIC) approaches, which can include screening for maternal Adverse Childhood Experiences (ACEs) during prenatal care. However, there is a paucity of research showing that TIC approaches are associated with improvements in maternal or offspring health outcomes. Using retrospective file review, the current study evaluated whether differences in pregnancy health and infant birth outcomes were observed from before to after the implementation of a TIC approach in a low-risk maternity clinic, serving women of low medical risk. Methods: Demographic and health data were extracted from the medical records of 601 women (*n* = 338 TIC care, *n* = 263 pre-TIC initiative) who received prenatal care at a low-risk maternity clinic. Cumulative risk scores for maternal pregnancy health and infant birth outcomes were completed by health professionals. Results: Using independent chi-squared tests, the proportion of women without pregnancy health risks did not differ for women from before to after the implementation of TIC, *χ*^2^ (2, 601) = 3.75, *p* = 0.15. Infants of mothers who received TIC were less likely to have a health risk at birth, *χ*^2^ (2, 519) = 6.17, *p* = 0.046. Conclusion: A TIC approach conveyed modest benefits for infant outcomes, but not maternal health in pregnancy. Future research examining other potential benefits of TIC approaches are needed including among women of high socio-demographic and medical risk.

## 1. Introduction

Over the last 20 years, it has become widely accepted that experiences of adversity in childhood, including abuse, neglect, and household dysfunction, set the stage for poor physical and mental health across the lifespan [1,2,3,4]. Exposure to adverse childhood experiences (ACEs) is an international problem, with two thirds of children exposed to some form of adverse experience prior to the age of 18 years worldwide [5]. This is particularly concerning as exposure to these experiences disrupt the core cognitive, affective, relational, and behavioral skills that are critical for mental and physical health [6]. Specifically, exposure to adverse childhood experiences is associated with chronic stress, often termed “toxic stress”, which can disrupt typical development through the expenditure of resources that would otherwise go towards healthy physical and emotional development [1]. Women and girls are disproportionately impacted by exposure to adversity, particularly sexual abuse and exposure to domestic violence [7,8]. These experiences have long term implications for their health [9] and mental health [10], as well as having consequences for fetal and infant health and development [11].

These high prevalence rates, and their subsequent impacts, have drawn attention to the need for trauma-informed approaches to women’s health, particularly in the maternity care setting. Specifically, approaches to care should take into consideration the high levels of trauma experienced by women in the general population and seek to mitigate its impact [12]. One such approach is trauma-informed care (TIC), which refers to services that recognize the impact of trauma on health and employ supportive practices that actively avoid re-traumatization and promote healing in individuals exposed to trauma [13]. A TIC approach includes many components and can include asking about or screening for adverse childhood experience in the context of primary care. While there is support for improvements in provider knowledge and acceptability with the implementation of TIC [14], to our knowledge little is known about the direct effect of TIC initiatives on maternal and infant health outcomes in the maternity setting. While TIC approaches paired with obtaining an ACE’s history have the potential to improve patient outcomes, there may also be risks associated with asking about past traumatic experiences during a brief primary care encounter [15]. Thus, the current study sought to evaluate whether differences in pregnancy health and infant birth outcomes were observed between groups of women receiving prenatal care at a low-risk maternity clinic (i.e., serving women at low risk for pregnancy or birth complications) prior to the implementation of a TIC initiative (standard care) and following the implementation of a TIC initiative (TIC group, Altona, Australia).

### 1.1. ACEs and Maternal Health Outcomes

While ACEs are associated with health difficulties across the lifespan, more recent literature has started to focus on the impact of ACEs on maternal-child health outcomes. Pregnancy and the postpartum period may be particularly vulnerable developmental stages for women who have experienced ACEs. Specifically, exposure to early life stress may have already resulted in biological impacts that are further exacerbated by the energy expensive process of reproduction, putting women exposed to ACEs at greater risk for poor health outcomes [16]. Indeed, exposure to ACEs is associated with increased risk for poor maternal health during pregnancy, including gestational diabetes, hypertension, and engaging in substance use [9,17,18,19]. Women with ACEs are also significantly more likely to experience mental health difficulties during their pregnancy [20]. Thus, the impact of ACEs on maternal health encompasses outcomes that are both biological and psychosocial in nature.

### 1.2. Maternal ACEs and Infant Birth Outcomes

The impact of maternal ACEs has been shown to extend beyond the prenatal period and is intergenerational in nature [10,11,17,21,22]. For instance, literature has shown that ACEs have strong implications for fetal immunological and neural development [19,23]. The epigenetic changes that occur as a result of maternal ACEs are theorized as being passed onto the developing fetus either directly through genetics, or indirectly through changes in the gestational environment [23,24]. These changes are related to an increased risk for poor infant birth outcomes including preterm birth and low birthweight [10,25,26]. Research has also shown that, for additional maternal ACE, there is an 18% increase in the likelihood of a child having a developmental delay [27]. Thus, maternal ACEs may have the potential to impact not just maternal health, but also an infant’s developmental trajectory from conception onwards.

### 1.3. Screening for ACEs in Prenatal Care

Catalyzed by the demonstrated association between ACEs and maternal and infant health outcomes, there has been a move to implement routine ACEs history taking (or screening) into the prenatal care setting [28,29]. This process involves asking a patient to self-report on their experiences of childhood abuse, neglect, and household dysfunction as part of their routine prenatal care. It is hypothesized that screening for ACEs presents an opportunity to identify mothers with experiences of early adversity and redirect them to resources to address ACEs and limit their intergenerational impact on health [29]. This could involve providing support to address parenting or maternal mental health concerns. Emerging studies have demonstrated that asking about ACEs within the prenatal care setting is feasible, and largely accepted by both physicians and patients [28,30].

Despite its feasibility, various concerns remain with widespread use of the ACEs questionnaire for screening purposes, especially in the absence of TIC approaches. The most notable concerns relate to the potential for patient re-traumatization, lack of availability of follow-up treatment, and the lack of specificity of the ACEs questionnaire [31,32]. Studies have shown that mothers with higher ACE scores tend to report less comfort with screening [28], and many physicians have raised the concern about a lack of training in trauma-informed care methods and knowledge of resources to best support patients found with high ACE scores [28,33,34]. With a lack of evidence that screening for social adversities, such as domestic violence, leads to an improvement in health outcomes [35,36,37], the question is raised as to whether the potential benefits of using the ACEs questionnaire outweighs the risks of potential harm.

### 1.4. Trauma-Informed Care and ACEs Screening in Pregnancy

The implementation of the ACEs questionnaire is best guided alongside the implementation of a trauma-informed care (TIC) approach [29,38]. The implementation of a TIC model would address most concerns with use of the ACEs questionnaire by ensuring that all aspects of care are committed to preventing harm and facilitating trauma recovery. A TIC approach to prenatal care is relational in nature, prioritizes compassionate care, and fosters environments that are emotionally safe, trustworthy, and transparent [12,39,40]. Training in TIC approaches includes using empathic listening, a non-judgmental stance, and behaviors that are open and transparent (e.g., a healthcare provider verbalizing what they are doing or why they are asking something). In the prenatal setting, the integration of trauma-informed care models could facilitate safe and supportive discussions about ACEs, as well as an assessment of social support and the safety of current relationships. Qualitative research with pregnant women and new mothers with trauma histories indicate that positive relationships, respect, and safety are key elements necessary in healthcare to reduce shame and mistrust that often accompany exposure to trauma [41,42], suggesting that TIC is an important component of asking about ACE histories.

While the need for trauma-informed care is well described [41,42], there is limited research evaluating its effectiveness in the prenatal care setting [43,44]. Sperlich et al. [39] conducted a review of trauma-informed approaches to maternity care and concluded that very few trauma-informed models and trauma-specific interventions have been developed for use with pregnancy and postpartum women. While there are trauma-informed perinatal programs (e.g., Minding the Baby [45], Survivor Mom’s Companion [46]) that have been designed and tested, these intensive programs are typically administered to targeted clinical populations and are not widely administered in primary care [39]. One study that did implement a TIC approach for pregnant women in primary care, which included an interview asking about past abuse and trauma as well as maternal mental health screening, found increases in attendance at prenatal appointments and decreased rates of preterm birth compared to rates prior to the implementation of the TIC approach [47]. However, no differences in infant birthweight, gestational age at birth, or number of premature infants were documented [47]. Identifying whether the implementation of a TIC approach that includes asking about trauma has implications for maternal and infant health outcomes as compared to standard care has important implications for maternity care practice.

### 1.5. Current Study

Using a retrospective file review, the goal of the current study was to evaluate whether the implementation of a trauma-informed care (TIC) approach in a low-risk maternity clinic was associated with differences in pregnancy health risks and infant birth outcomes, from pre- to post-implementation of the TIC approach. Given that TIC approaches have demonstrated limited improvements in mental health outcomes in outpatient settings [48], we expect that there would be little to no difference in maternal health outcomes for individuals who received TIC versus standard care in the prenatal care setting. However, based on one research study demonstrating a reduction in the number of low-birth weight infants after the implementation of a TIC initiative [47], we anticipate a potential improvement in infant birth outcome with the implementation of a TIC approach.

## 2. Materials and Methods

### 2.1. Participants

The sample was a non-probabilistic convenience sample of 601 women receiving prenatal care at a low-risk maternity clinic in a large, urban center in Western Canada. The maternity clinic is comprised of a multi-disciplinary team of primary care physicians, nurses, social workers and mental health consultants, providing prenatal care to over 2000 women annually. New patients are referred to the clinic at approximately 18–20 weeks gestation by their family physician if they meet the following criteria: singleton pregnancy, intend to have a vaginal birth, <42 years of age, have a pre-pregnancy body mass index <40, have no major fetal abnormalities or uterine anomalies, do not have any pre-existing health concerns (e.g., autoimmune disease), and have not had significant complications during previous pregnancies (e.g., preterm birth, stillbirth). On average, women attended 9.07 (SD = 3.31) prenatal appointments with the number of appointments ranging from 1 to 16.

### 2.2. Study Design

Using a retrospective file review [49], women were categorized as being in one of two groups: (1) the standard care group, which included 263 women who received care at the clinic from May 2016–2017 which was prior to the implementation of the TIC initiative; and (2) the TIC group, which included 338 women who received care between June 2017 and December 2018 after the TIC initiative and ACEs screening were implemented. Consistent with TIC practices of giving patients choice in their own care, completing the ACEs questionnaire was optional. Approximately 86% agreed to complete the ACEs questionnaire and only patients who did so were included in the TIC group. Women who did not complete the ACEs questionnaire after the implementation of the TIC approach were not included as they would not have had an explicit conversation about trauma with their provider, which is considered to be an active part of the initiative.

#### 2.2.1. Data Collection Approach

Demographic information, pregnancy health risks and infant birth outcomes were collected from the electronic medical record by four independent research assistants and de-identified. In line with guidelines for file review methodology [49], two research assistants piloted data collection and coding protocols, and two additional research assistants were trained on extraction. To ensure reliability in extraction, 20% of patient files were double coded. Intraclass correlation for the antepartum risk assessment score, used to derive the outcomes in the current study, was 0.78.

#### 2.2.2. Implementation of the TIC Initiative

The trauma-informed care (TIC) initiative implemented at the prenatal care practice was implemented in June 2017. The TIC initiative was multi-pronged, including the identification and training of a peer champion, the training of primary care staff and physicians, patient awareness, and the implementation of standardized screening for childhood trauma and mental health symptoms, as well as referrals to mental health services as needed [50]. First, a physician champion from the maternity clinic was trained in trauma-informed approaches as well as change management processes related to implementing a TIC initiative [13]. The physician champion was selected through their role as medical lead of the low-risk obstetrics program for the area. A physician was selected as the peer champion, as the majority of practitioners who were asking about ACEs in the prenatal care setting were physicians, and shared background and training were thought to contribute to increased uptake. The physical champion attended a two-day conference and workshop on trauma-informed care and change management. The physician champion also completed an online TIC course module provided by the local health authority [51]. After receiving training, the physician champion followed the Awareness, Desire, Knowledge, Ability and Reinforcement (ADKAR) model for change management to coordinate training and the preparation of resources [52].

Next, the peer-physician champion provided continuing education in the form of a retreat, two lunch-and-learns, and online resources about TIC and implementing ACEs screening into their practice. These training sessions were not mandatory, but were well attended by the majority of staff. The TIC continuing education focused on understanding the biopsychosocial impact of trauma on health and cultivating empathy for patients who have experienced trauma. Skill building exercises that focused on motivational interviewing techniques, such as active listening and empathic responding, were discussed. Training and informational materials on how to ask patients about trauma using a screening questionnaire and how to provide follow-up were also shared. The training materials were developed by the physician champion and were informed by resources provided by the Substance Abuse and Mental Health Services Administration [13] as well as the Alberta Family Wellness Initiative Brain Story Certification [53]. Next, patient awareness of trauma and mental health was fostered, including posters in the clinic, brochures, and information on the clinic website. The last step was to implement the ACEs history taking. During the first week that the history taking was rolled out, the physician champion was available to consult, answer questions, and provide peer support.

#### 2.2.3. Implementation of ACEs Screening

With regards to screening for ACEs, patients were provided with a handout at the second prenatal visit explaining the rationale for asking about childhood adversity. Subsequently, patients voluntarily completed a modified version of the 10-item ACEs questionnaire which included the following items: emotional abuse, physical abuse, sexual abuse, emotional neglect, physical neglect, parental loss, separation or divorce, exposure to domestic violence, household substance abuse difficulties, household member with mental illness, or an incarcerated household member [54]. Only the total score was shared with the practitioner. The results of the ACEs questionnaire were subsequently reviewed with the physician, along with the perceived impact of any endorsed items and a discussion about current mental health needs and available supports. The sum ACEs score (i.e., total number endorsed as having occurred) was recorded on the health record. Referrals for follow-up care for mental health support were provided as needed. All patients also received an information handout with online and local mental health resources. Following these discussions, 18 women (5.3%) received a referral to a therapist or mental health clinician after ACEs screening.

### 2.3. Measures

#### 2.3.1. Demographic Characteristics

Patient demographic characteristics were extracted from the electronic medical record. This information was collected via maternal self-report and recorded in the medical record by a healthcare provider. Maternal age in years, being racially/ethnically minoritized (i.e., not from the dominant group), marital status (single versus married/common-law), primiparous (yes/no), and the presence of financial stress (yes/no) were extracted. Mothers were also asked to provide their current occupation. From the occupation provided, occupational prestige was coded using the Hollingshead Four Factor Index of Social Status [55]. The reliability coding for occupational prestige had intraclass correlation coefficients that ranged from 0.93 to 0.99, indicating high reliability.

#### 2.3.2. Pregnancy Health Risk

Pregnancy heath risk was operationalized using items from the antepartum risk assessment score (ARA; [56]), which is a risk assessment tool completed by a health professional, used on the prenatal record to evaluate each woman’s medical risk of birth complications [57]. Items are endorsed as either having occurred (score of 1) or not occurred (score of 0). Given the low frequency of some of the individual items, a cumulative pregnancy health risk score of nine total items was derived based on items from the ARA including polyhydramnios/oligohydramnios, bleeding in pregnancy, gestational hypertension, gestational diabetes, anemia, placenta previa, tobacco use in pregnancy, alcohol use in pregnancy, or drug use in pregnancy. 

#### 2.3.3. Infant Birth Outcomes

In line with previous literature, cumulative risk scores were calculated for infant birth outcomes [11,17]. Each birth outcome was given a score of 0 (not present) or 1 (present) and summed to create a total risk score. The cumulative infant birth outcomes score included the following 13 items: preterm birth, stillbirth, Apgar score < 7, low birth weight (<2500 g), neonatal intensive care stay, head abnormality, abdomen abnormality, genetic screening abnormality, musculoskeletal abnormality, genital/rectal abnormality, and central nervous system abnormality. 

### 2.4. Analytic Plan

Descriptive statistics were estimated for all continuous and categorical variables, including participants’ demographic characteristics, maternal health risk, and infant birth outcomes scores. Chi-squared and *t*-test statistics were used to compare demographic variables between the standard care group and the TIC group to identify potential co-variates. Differences in the proportion of women with 0, 1, and 2 or more pregnancy and birth health risks (trimmed due to the small cell counts for those with greater than two health risks) were examined using independent chi-squared tests. Statistical significance for all tests were set at *p* < 0.05. All analyses were conducted using SPSS statistics version 25.

#### 2.4.1. Sample Size Calculation

An a priori sample size calculation was conducted in order to ensure there was adequate power for conducting the analyses. In order to detect a small to moderate effect size (d = 0.30) with power of 0.95 and an error probability of 0.05, it was determined that a minimum total sample size of 492, with 277 and 215 participants respectively in each group, would be needed. Therefore, the current total sample size of 601 was deemed adequate.

#### 2.4.2. Missingness Analyses

There was no missing data for the maternal health risk outcome, but 82 (13.6%) participants had missing data for the infant birth outcome. Analyses to determine whether there were demographic differences for individuals with and without missing data were conducted using t-tests and chi-squared tests. There were no mean differences in maternal age (t = −0.74, *p* = 0.46) or maternal occupational prestige (t = 0.65, *p* = 0.52). There was also no difference in the proportion of mothers who were primiparous (χ^2^ = 1.61, *p* = 0.21), minoritized (χ^2^ = 1.73, *p* = 0.19), single (χ^2^ = 0.59, *p* = 0.44), or had financial stress (χ^2^ = 0.14, *p* = 0.71). Thus, a complete case approach was used for the infant birth outcome analyses.

## 3. Results

### 3.1. Descriptive Statistics

Overall demographic characteristics as well as those for the standard care group and the TIC group are presented in Table 1. Across both groups (N = 601), mothers had an average age of 30.95 years (SD = 4.45) and 43.4% (261/601) identified as a racially minoritized. There were no significant differences in demographic characteristics between the TIC and standard care groups. Thus, no demographic characteristics were used as covariates. With regards to results from the ACEs screening, 16.0% of women reported one ACE, 10.0% reported two ACEs, 2.4% reported three ACEs, and 3.8% reported four or more. That is, 32.3% of women reported at least one ACE.

### 3.2. Maternal Health Risk and Infant Birth Outcomes

Descriptive statistics for the maternal health risk score and infant birth outcome score are presented in Table 2. The frequency of individual items for both cumulative scores are presented in Appendix A Table A1. In line with the low-risk nature of the maternity clinic, 37% of women in the TIC group and 33.8% of women in the standard care group had at least one maternal health risk in pregnancy. Scores ranged from 0–4. For the infant birth outcomes, 26.9% of women in the TIC group and 29.3% of women in the standard care group had infants with at least one adverse birth outcome. The mean score for the maternal health risk variable was 0.48 (SD = 0.72) and the mean score for the cumulative infant birth outcomes was 0.45 (SD = 0.77).

### 3.3. Differences in Maternal Health Risk in Pregnancy for the TIC versus Standard Care

Using an independent chi-squared test, difference in the proportion of women who had 0, 1, or 2+ health risks in pregnancy in the TIC and SC groups were compared (See Table 2). The number of maternal health risks were combined to be evaluated across three groups due to small cell sizes (i.e., only 2.2% (*n* = 13) of women had more than 2 health risks in pregnancy). The proportion of women without pregnancy health risks did not differ for women who received TIC versus SC, *χ*^2^ (2, 601) = 3.75, *p* = 0.15.

### 3.4. Differences in Infant Birth Outcomes for the TIC versus Standard Care Group

An independent chi-squared test was used to examine differences in infant birth outcomes for those in the TIC versus standard care groups (0, 1, 2+ adverse outcomes). The number of infant adverse birth outcomes were combined to be evaluated across three groups due to small cell sizes (i.e., only 3.3% (*n* = 17) of women had 3 or more had poor infant birth outcomes). Infants of mothers who received TIC were less likely to have an adverse birth outcome than infants of women in the SC group, *χ*^2^ (2, 519) = 6.17, *p* = 0.046. Specifically, 70.45% of infants in the TIC group had no birth health risks, whereas only 63.5% of infants in the SD group had no birth health risks.

## 4. Discussion

The goal of the current study was to evaluate whether the implementation of a trauma-informed care (TIC) approach, which included asking about maternal ACEs and maternal mental health in a low-risk maternity clinic, was associated with differences in pregnancy health and infant birth outcomes, for better or for worse, as compared to standard care. In the current study we found that asking about ACEs in the prenatal care setting in the context of a broader TIC initiative as compared to standard care was not associated with differences in maternal pregnancy health and only modestly associated with better infant birth outcomes. Notably, the TIC initiative was not associated with increases in pregnancy or birth risks. Below we discuss implications of our findings and directions for future research.

One conclusion that is frequently drawn from studies identifying an association between ACEs and adverse health is that asking or screening for ACEs will provide important information to a healthcare provider that will ultimately improve maternal and infant health outcomes. Specifically, that knowing about a mother’s ACE score could initiate a referral for support and services that will improve outcomes. Indeed, in simulation modelling of ACEs screening, it has been suggested that asking about ACEs in primary care would be associated with an increase in the use of mental health interventions [58] Feeling supported, understood, and cared for by a healthcare provider could also be associated with reductions in stress or affect regulation difficulties in pregnancy [39], which could subsequently lead to improved maternal health and birth outcomes. There have also been arguments to the contrary, suggesting that routinely asking about trauma could lead to re-traumatization and increased distress [15,31,59,60]. Although trauma-informed care initiatives have demonstrated improvements in practitioners’ knowledge, practice, and collaboration [61], there is limited evidence that it improves patient outcomes [48]. One exception is a study comparing birth outcomes before and after the implementation of a TIC initiative with teen mothers in a maternity care setting, which demonstrated reductions in the percentage of low birthweight infants, but no differences in gestational age at birth, prematurity, or mean birthweight [47]. Thus, our findings that a TIC approach was associated with a modest reduction in infant birth complications is in line with previous research demonstrating benefits of the approach.

There are several potential reasons why the implementation of ACEs screening as part of a broader Trauma-Informed approach was not associated with improvements in maternal pregnancy health. First, while healthcare providers received training in TIC approaches, implementation of the ACEs screening questionnaire, and the provision of resource sheet or referrals as needed, only 5% of women received an external referral to a mental health clinician. Women in both groups had access to a behavioral health consultant for psychosocial support, but the extent and duration of this support was limited. Separate research on screening for domestic violence has shown that, in the absence of an intervention, no changes or reductions in domestic violence are observed [36]. Therefore, while TIC training and obtaining an ACEs history may be a first step, it may be insufficient on its own to have immediate effects on maternal health outcomes. However, in the current study we did observe modest downstream effects of the TIC approach to infant outcomes.

Similarly, the dose of TIC in the current study may have been insufficient to lead to improvements in maternal health outcomes. Although maternity care providers at the current clinic received training, participated in workshops, and had access to an expert champion to consult, the application of the TIC approach did not extend beyond the clinic to other settings in the healthcare setting that women access (e.g., blood test laboratories, diagnostic imaging centers, prenatal classes, or labor and delivery hospital units). It may be that a larger system-wide approach to TIC is needed in order to have a demonstrated impact on maternal health outcomes.

Another reason for our lack of findings for maternal health in the current study may be related to the lack of diversity with regards to sociodemographic risk (i.e., poverty, low education) and medical risk. Women in the current study were not at high risk of pregnancy or birth complications, which may have reduced our ability to detect an effect with regards to these outcomes. The majority of women were not experiencing financial stress and had adequate household family incomes. This sample represents an urban, racially and ethnically diverse group of women with access to universal healthcare and other government-funded social safety nets. An important direction for future research should be to investigate changes in maternal health in more medically and socio-demographically diverse samples.

As mentioned previously, both health care providers and patients have found the implementation of a TIC approach and conversations about experiences in childhood to be feasible, acceptable, and helpful [28,50]. While differences in maternal health outcomes were not documented in the current study, it may be that the noted benefits or improvements to the patient–provider relationship or the sense of understanding that patients felt was not documented here. For example, anecdotally, providers have shared that they booked additional time for appointments with clients who had significant trauma histories. It will be important for future research to expand the measurement of outcomes in order to capture the broader outcomes that may have shifted as a result of the TIC initiative.

### Limitations

There are limitations to the current study that warrant mention. First, the current study employed retrospective file review methodology and was observational in nature, thus, we are limited in our ability to make causal conclusions about these findings. Future research would benefit from employing a randomized-controlled trial methodology in order to make more firm conclusions about differences among groups. Second, the current study was conducted in a low-risk maternity clinic. Although our findings generalize to the majority of pregnant people who do not experience severe complications in their pregnancy, our sample did not include individuals who are at most elevated risk of experiencing pregnancy complications or adverse birth outcomes. The number of women reporting at least one ACE in the current sample was also lower than in other samples [54], which may be attributed to the low-risk nature of the clinic or lack of comfort with disclosing directly in a screening context [62]. Additionally, 14% of women declined to complete the ACEs questionnaire, who could have been patients with the highest levels of ACEs. Similar research in high-risk obstetrical clinics is needed to generalize findings to that population, specifically. Lastly, although the TIC approach that was initiated at the maternity clinic in the current study was universally implemented within the clinic, there were no fidelity checks conducted to ensure the use of the approach was maintained across 40 care providers. Additionally, we are unable to tease out whether the effects observed were due to the TIC training, the implementation of ACEs screening, or the implementation of maternal mental health screening. Future research should evaluate the fidelity of the TIC implementation.

## 5. Conclusions

The current study found that the implementation of a TIC approach, which included asking about maternal ACEs and screening for maternal mental health difficulties, was not associated with differences in maternal pregnancy health and was associated with modest improvement in infant birth outcomes in a low-risk maternity clinic. Although previous research has demonstrated that a TIC approach in the maternity care setting is associated with improvements in practitioner knowledge and there is a high level of acceptability among patients and care providers [14], we provide novel evidence that TIC approaches may have intergenerational benefits. Expanding the repertoire of possible outcomes associated with TIC approaches, such as maternal mental health, pregnancy stress, and perceived social support, is an important future research direction. Rigorous evaluations of TIC approaches in the maternity setting that move beyond retrospective file review, self-report, and provider perspectives are also needed.

## Figures and Tables

**Table 1 children-08-01061-t001:** Demographic characteristics for the trauma-informed care (TIC) versus standard care groups.

Characteristic	Overall	TIC Group (*n* = 338)	Standard Care (*n* = 263)	t-Value or χ^2^	*p*-Value
Maternal age (years), M (SD)	30.95 (4.45)	30.91 (4.24)	31.01 (4.71)	0.282	0.78
Occupational prestige, M (SD)	4.44 (3.28)	4.59 (3.29)	4.23 (3.27)	−1.243	0.21
Race and Ethnicity, N (%)					
Minority	261 (43.4)	151 (47.8)	110 (46.4)	0.102	0.75
Non-minority	292 (48.6)	165 (52.2)	127 (53.6)		
Marital Status, N (%)				0.57	0.45
Single	27 (4.5)	17 (5.3)	10 (4.0)		
Married/Common-law	547 (91.0)	304 (94.7)	243 (96.0)		
Primiparous, N (%)				0.28	0.59
Yes	328 (54.6)	181 (53.7)	147 (55.9)		
No	272 (45.3)	156 (46.3)	116 (44.1)		
Financial Stress, N (%)				0.25	0.62
Yes	35 (5.8)	18 (5.6)	17 (6.5)		
No	549 (91.3)	306 (94.4)	243 (93.5)		

**Table 2 children-08-01061-t002:** Differences in Mean Cumulative Risk Scores for TIC versus Standard Care Groups.

	TIC Group (*n* = 338)	Standard Care (*n* = 263)		
Cumulative Risk Score	N (%)	N (%)	χ^2^	*p*-Value
Maternal Health Risk				
0	213 (63.02)	174 (66.16)	3.75	0.15
1	91 (26.92)	74 (28.14)		
2+	34 (10.06)	15 (5.70)		
Infant Health Outcomes				
0	217 (70.45)	134 (63.5)	6.17	0.046
1	62 (20.13)	62 (29.38)		
2+	29 (9.41)	15 (7.1)		

## Data Availability

Data for the current project cannot be made publicly available for sharing.

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
