# Peer review of "Maternal-Child Health Outcomes from Pre- to Post-Implementation of a Trauma-Informed Care Initiative in the Prenatal Care Setting: A Retrospective Study"

_children, 2021, doi:10.3390/children8111061_

Round 1
Reviewer 1 Report
Thank you for the opportunity to review this manuscript. Strengths include a focus on an important yet understudied health issue (trauma informed care and maternal-child health outcomes), a relatively large sample size, and setting in the “real world.” Opportunities for refinement are detailed below. Of note, the paper is missing a discussion and conclusion, which I assume is a document error (?), so I could not complete my review.
*Page 1 line 34 – The authors note that “as exposure to these experiences disrupt the core cognitive, affective, relational, and behavioural skills.” Can they add more context as to why? One or two sentences about pathways underlying these associations (e.g., effects of chronic/severe stress) could be helpful to the unfamiliar reader.
*Page 1 line 37 – The authors may want to acknowledge that women and girls are more impacted not only by sexual abuse, but also by intimate partner violence.
*Page 2 line 37 – The authors may want to clarify that TIC is about not only avoiding re-traumatization, but also promoting healing.
*Page 3 line 140 – This paragraph is a bit hard to follow. For example, as currently written it is a bit hard to parse whether it is a one group pre-post or intervention/control design. Can the authors streamline? (E.g., can they more explicitly address the study design, aims, and hypotheses)? I recognize this is addressed later in the methods, but clarity here would help.
*Page 4 line 191 – Much more detail about the intervention is needed. Who developed the training materials, and were they adapted from existing sources? What was the dose/duration of the components of the trainings, such as the physician training? Who was eligible to serve as a “peer champion”? How often were “lunch and learns” and were they mandatory?
*Were any process or fidelity data collected? For example, how often did the women in the TIC group interact with the team? Was it only during the one visit, or did most have multiple visits during the time period.
*Page 5 line 212 – Can the authors lists the 10 ACEs? Given the variation in ACEs assessment across the literature, explicit detail would be helpful.
*Page 6 line 277 – What are the authors thoughts on why on ~1/3 of mothers reported an ACE? That is much lower than estimates in USA; perhaps it is consistent with Canadian data but I am unsure.
*The paper is missing a discussion and conclusion section.
Author Response
Manuscript ID: children-1426479
Title: Do maternal-child health outcomes differ for women who receive care after the implementation of a trauma-informed care initiative in the prenatal care setting?
Thank you for the opportunity to submit our revised manuscript and to the reviewers for their thoughtful comments. We provide responses to each of the reviewers below and have made the changes using track changes in the word document.
Reviewer 1:
Thank you for the opportunity to review this manuscript. Strengths include a focus on an important yet understudied health issue (trauma informed care and maternal-child health outcomes), a relatively large sample size, and setting in the “real world.” Opportunities for refinement are detailed below. Of note, the paper is missing a discussion and conclusion, which I assume is a document error (?), so I could not complete my review.
Response: We are grateful to the reviewer for their time in providing feedback on the manuscript. We apologize for the document error, which occurred (unbeknownst to us) during the document uploading process. We have now included a version of the manuscript with our original discussion and conclusion section.
- Page 1 line 34 – The authors note that “as exposure to these experiences disrupt the core cognitive, affective, relational, and behavioural skills.” Can they add more context as to why? One or two sentences about pathways underlying these associations (e.g., effects of chronic/severe stress) could be helpful to the unfamiliar reader.
Response: As requested by the reviewer, we have added the following sentence on page 1 lines 37-40 to contextualize how ACEs disrupt development: “Specifically, exposure to adverse childhood experiences are associated with chronic stress, often termed “toxic stress”, which can disrupt typical development through the expenditure of resources that would otherwise go towards healthy physical and emotional development [1].”
- Page 1 line 37 – The authors may want to acknowledge that women and girls are more impacted not only by sexual abuse, but also by intimate partner violence.
Response: We agree that intimate partner violence as an exposure is important and have added it on page 1, line 41-42: “Women and girls are disproportionately impacted by exposure to adversity, particularly more severe forms such as sexual abuse and exposure to domestic violence [7,8].”
- Page 2 line 37 – The authors may want to clarify that TIC is about not only avoiding re-traumatization, but also promoting healing.
Response: We agree with this addition and as such have rephrased our definition of TIC to include a focus on healing on Page 2, lines 58-60: “One such approach is trauma-informed care (TIC), which refers to services that understand the impact of trauma on health and employ supportive practices that actively avoid re-traumatization and promote healing in individuals exposed to trauma [13].”
- Page 3 line 140 – This paragraph is a bit hard to follow. For example, as currently written it is a bit hard to parse whether it is a one group pre-post or intervention/control design. Can the authors streamline? (E.g., can they more explicitly address the study design, aims, and hypotheses)? I recognize this is addressed later in the methods, but clarity here would help.
Response: We agree with the reviewer that streamlining the paragraph describing the current study was needed as details on the methods are provided later in the paper. We have streamlined the paragraph on page 4, lines 154-163 as follows: “Using a retrospective pre-post design, the goal of the current study was to evaluate whether the implementation of a trauma-informed care (TIC) approach in a low-risk maternity clinic was associated with differences in pregnancy health risks and infant birth outcomes, from pre- to post-implementation of the TIC approach. Given that TIC approaches have demonstrated limited improvements in mental health outcomes in outpatient settings [48], we expect that there would be little to no difference in maternal health outcomes for individuals who received TIC versus standard care in the prenatal care setting. However, based on one research study demonstrating a reduction in the number of low-birth weight infants after the implementation of a TIC initiative [47], we anticipate a potential improvement infant birth outcomes with the implementation of a TIC approach.”
- Page 4 line 191 – Much more detail about the intervention is needed. Who developed the training materials, and were they adapted from existing sources? What was the dose/duration of the "components of the trainings, such as the physician training? Who was eligible to serve as a “peer champion”? How often were “lunch and learns” and were they mandatory?
Response: We appreciate the need to add more detail with regards to the TIC initiative that was implemented at the clinic. As such we have added the additional information requested and appropriate references on page 5 lines 227-252: “The physician champion was selected through their role as medical lead of the low-risk obstetrics program for the area. A physician was selected as the peer champion as the majority of practitioners who were asking about ACEs in the prenatal care setting were physicians and shared background and training were thought to contribute to increased uptake. The physical champion attended a two-day conference and workshop on trauma-informed care and change management. The physician champion also completed an online TIC course module provided by the local health authority[51]. After receiving training, the physician champion followed the Awareness, Desire, Knowledge, Ability and Reinforcement (ADKAR) model for change management to coordinate training and the preparation of resources [52].
Next, the peer-physician champion provided continuing education in the form of a retreat, two lunch-and-learns, and online resources about TIC and implementing ACEs screening into their practice. These trainings were not mandatory, however, were well attended by the majority of staff. The TIC continuing education focused on understanding the biopsychosocial impact of trauma on health and cultivating empathy for patients who have experienced trauma. Skill building exercises that focused on motivational interviewing techniques, such as active listening and empathic responding, were discussed. Training and informational materials on how to ask patients about trauma using a screening questionnaire and how to provide follow-up were also shared. The training materials were developed by the physician champion and were informed by resources provided by the Substance Abuse and Mental Health Services Administration [13] as well as the Alberta Family Wellness Initiative Brain Story Certification [53]. Next, patient awareness about trauma and mental health were created, including posters in the clinic, brochures, and information on the clinic website. The last step was to implement the ACEs history taking. The first week that the history taking was rolled out, the physician champion was available to consult, answer questions, and provide peer support.”
- Were any process or fidelity data collected? For example, how often did the women in the TIC group interact with the team? Was it only during the one visit, or did most have multiple visits during the time period.
Response: The reviewer raises an important question with regards to fidelity checks and unfortunately this is a limitation of the current study which we acknowledge in the limitations section on page 10, lines 472-474: “Lastly, although the TIC approach that was initiated at the maternity clinic in the current study was universally implemented within the clinic, there were no fidelity checks that were conducted to ensure the use of the approach was maintained across 40 care providers.”
With regards to the dose or number of times that patients interacted with the team, this number would have varied by the number of appointments attended, which was on average 9 appointments. We provide this number in the manuscript on page 4, lines 175-177.
- Page 5 line 212 – Can the authors lists the 10 ACEs? Given the variation in ACEs assessment across the literature, explicit detail would be helpful.
Response: We have added the list of the 10 ACEs items on page 5, lines 257-260: “Subsequently, patients voluntarily completed a modified version of the 10-item ACEs questionnaire which included the following items: emotional abuse, physical abuse, sexual abuse, emotional neglect, physical neglect, parental loss, separation or divorce, exposure to domestic violence, household substance abuse difficulties, household member with mental illness, or an incarcerated household member [51].”
- Page 6 line 277 – What are the authors thoughts on why on ~1/3 of mothers reported an ACE? That is much lower than estimates in USA; perhaps it is consistent with Canadian data but I am unsure.
Response: We agree that discussing the lower ACEs reporting in the current study provides important context. As such we have expanded on this discussion in the limitations section on p. 10, lines 466-468: “The number of women reporting at least one ACE in the current sample was also lower than in other samples [54], which may be attributed to the low-risk nature of the clinic or lack of comfort with disclosing directly in a screening context [65].”
- The paper is missing a discussion and conclusion section.
Response: We apologize for the document error and have now included a version with the discussion and conclusion section.
Reviewer 2 Report
Thank you for the possibility to review this interesting study about the implementation of trauma-informed care initiative in prenatal settings. The idea of the study is great, however, some methodological clarifications are needed. Starting from the title, the authors should make clear that the study is totally retrospective. The design should be explicit in the abstract. The text in the introduction is flowing nicely. Should the paragraph 1.5. Current study, be placed in the methods section? In the methods, more details are needed. What was the justification for the sample size? If the clinic covers over 2000 pregnancies in a year, how were the study women selected? Did the women completing the ACE questionnaire know that they were included in the study? If yes, was the informed consent obtained? I would suggest using a separate sub-title for the description of the intervention implementation. Did the authors consult a statistician with the analysis? Why were so many infant outcomes missing?
About the results. The main results are described well. Please consider combining the tables 2 and 3. Maybe some more complicated analyses could also be considered? What about the 32,3% of women who reported some ACEs?
My main concern about the study is that the discussion part (including limitations and conclusions) is totally missing. What is the novelty of this study?
Author Response
Manuscript ID: children-1426479
Title: Do maternal-child health outcomes differ for women who receive care after the implementation of a trauma-informed care initiative in the prenatal care setting?
Thank you for the opportunity to submit our revised manuscript and to the reviewers for their thoughtful comments. We provide responses to each of the reviewers below and have made the changes using track changes in the word document.
Reviewer 2:
Thank you for the possibility to review this interesting study about the implementation of trauma-informed care initiative in prenatal settings. The idea of the study is great, however, some methodological clarifications are needed.
Response: We appreciate your time in reviewing our manuscript and providing feedback.
- Starting from the title, the authors should make clear that the study is totally retrospective.
Response: We appreciate the need to be explicit with regards to the study design in the title and have changed it to: “Maternal-child health outcomes from pre- to post-implementation of a trauma-informed care initiative in the prenatal care setting: A retrospective study”.
- The design should be explicit in the abstract.
Response: To address the reviewer’s request, we have added the study design to the abstract on page 1, lines 15-17: “Using a pre-post design, this retrospective study evaluated whether differences in pregnancy health and infant birth outcomes were observed from before to after the implementation of a TIC approach in a low-risk maternity clinic.”
- The text in the introduction is flowing nicely. Should the paragraph 1.5. Current study, be placed in the methods section?
Response: Thank you for your feedback. In line with comments from Reviewer 1, we have streamlined section 1.5 and included details about the methods in the methods section.
- In the methods, more details are needed. What was the justification for the sample size?
Response: As requested by the reviewer we have added a section on Page 6, lines 313-318 detailing our justification of sample size: “An a priori sample size calculation was conducted in order to ensure there was adequate power for conducting the analyses. In order to detect a small to moderate effect size (d=0.30) with power of 0.95 and an error probability of 0.05, it was determined that a minimum total sample size of 492, with 277 and 215 participants respectively in each group would be needed. Therefore, the current total sample size of 601 was deemed adequate.”
- If the clinic covers over 2000 pregnancies in a year, how were the study women selected?
Response: As detailed on page 4, lines 166-168, the current sample “was a non-probabilistic convenience sample of 601 women receiving prenatal care at a low-risk maternity clinic in a large, urban centre in Western Canada”. Women were included from the start of the implementation of the TIC approach in June 2017 and data collection stopped when an adequate sample size was obtained.
- Did the women completing the ACE questionnaire know that they were included in the study? If yes, was the informed consent obtained?
Response: The current study was a retrospective file review of archival data and as such a waiver of consent after the fact was obtained. We did receive ethics approval from the Institutional Review Board to conduct this study. Details of the consent waiver and ethics review board approval are provided on page 11, lines 511-515:
Institutional Review Board Statement: The study was conducted according to the guidelines of the Declaration of Helsinki, and approved by the Institutional Review Board of the University of Calgary (REB18-1601) on January 7th, 2019.
Informed Consent Statement: Patient consent was waived due to the archival nature of the data and infeasibility of retro-actively obtaining patient consent.
- I would suggest using a separate sub-title for the description of the intervention implementation.
Response: We agree with the reviewer and have added a separate sub-title for the implementation of the TIC initiative on Page 4, line 200.
- Did the authors consult a statistician with the analysis? Why were so many infant outcomes missing?
Response: We thank the author for their consideration to the missing infant outcomes. Both the lead and senior authors of the current paper have extensive methodological skills in research methods and analysis, as well as content expertise in maternal mental health, birth outcomes, and child development. Additionally, our co-author, Dr. Sheila McDonald, holds a PhD in Epidemiology and Biostatistics with a specific focus on life course epidemiology and causal inference. Taken together, we have been thoughtful about our statistical approach to the current analysis.
With regards to the missing data for the infant outcomes, we conducted a missing data analysis on page 6 and 7, lines 319-327 and did not find any systematic associations for missing data. One reason might be that the records of mothers who returned directly for follow-up within the community were not sent to the maternity clinic. However, this would not introduce bias to the current study as this would be a random occurrence.
- About the results. The main results are described well. Please consider combining the tables 2 and 3. Maybe some more complicated analyses could also be considered? What about the 32,3% of women who reported some ACEs?
Response: We appreciate the reviewer’s comments. As requested, we have combined Tables 2 and 3 for simplicity. With regards to additional analyses, the goal of the current study was to explicitly evaluate the impact of the TIC initiative. We have explored the association between ACEs and maternal and child health outcomes in another manuscript that is currently under review and have not included them here to avoid overlap.
- My main concern about the study is that the discussion part (including limitations and conclusions) is totally missing. What is the novelty of this study?
Response: We apologize for the document error and have now included a version with the discussion and conclusion section. We specifically outline the novelty of the study on page 10, lines 484-488: “Although previous research has demonstrated that a TIC approach in the maternity care setting is associated with improvements in practitioner knowledge and there is a high level of acceptability among patients and care providers [14], we provide novel evidence that TIC approaches may have intergenerational benefits.”
Round 2
Reviewer 2 Report
Thanks for the authors for their comprehensive work with the previous comments. I have only some minor comments. First, I'm confused with "the retrospective pre- and post design". In the Study design chapter the authors use "retrospective file review" which is better I think. Second, I would start the discussion section with the results of this study, not previous results.
